# Role of Drug Transporters in Elucidating Inter-Individual Variability in Pediatric Chemotherapy-Related Toxicities and Response

**DOI:** 10.3390/ph15080990

**Published:** 2022-08-11

**Authors:** Ashwin Kamath, Suresh Kumar Srinivasamurthy, Mukta N. Chowta, Sheetal D. Ullal, Youssef Daali, Uppugunduri S. Chakradhara Rao

**Affiliations:** 1Department of Pharmacology, Kasturba Medical College, Mangalore, Manipal Academy of Higher Education, Manipal 575001, India; 2Department of Pharmacology, Ras Al Khaimah College of Medical Sciences, Ras Al Khaimah Medical and Health Sciences University, Ras Al Khaimah P.O. Box 11172, United Arab Emirates; 3Department of Anaesthesiology, Pharmacology, Intensive Care and Emergency Medicine, Division of Clinical Pharmacology and Toxicology, Geneva University Hospitals, 1205 Geneva, Switzerland; 4CANSEARCH Research Platform in Pediatric Oncology and Hematology, Department of Pediatrics, Gynecology and Obstetrics, University of Geneva, 1205 Geneva, Switzerland

**Keywords:** chemotherapy, ontogeny, maturation, treatment response, toxicity, children, pediatric, efflux, influx, drug levels, pharmacogenetics

## Abstract

Pediatric cancer treatment has evolved significantly in recent decades. The implementation of risk stratification strategies and the selection of evidence-based chemotherapy combinations have improved survival outcomes. However, there is large interindividual variability in terms of chemotherapy-related toxicities and, sometimes, the response among this population. This variability is partly attributed to the functional variability of drug-metabolizing enzymes (DME) and drug transporters (DTS) involved in the process of absorption, distribution, metabolism and excretion (ADME). The DTS, being ubiquitous, affects drug disposition across membranes and has relevance in determining chemotherapy response in pediatric cancer patients. Among the factors affecting DTS function, ontogeny or maturation is important in the pediatric population. In this narrative review, we describe the role of drug uptake/efflux transporters in defining pediatric chemotherapy-treatment-related toxicities and responses. Developmental differences in DTS and the consequent implications are also briefly discussed for the most commonly used chemotherapeutic drugs in the pediatric population.

## 1. Introduction

In recent decades, strategies involving risk stratification and multimodal treatments have led to significant improvements in response rates to chemotherapy in pediatric oncology ranging from leukemias to brain cancers [1]. However, there is still large scope for minimizing adverse drug reactions and improving the prognosis for various pediatric cancers. Precisely targeted therapies are proposed to individualize treatments by diminishing morbidity and mortality [2]. Cancer chemotherapy outcomes are affected by the inter-individual variability in the pharmacokinetics or dynamics of the drugs, mainly owing to the genetic polymorphisms altering the amount or function of gene products, i.e., proteins. These proteins include drug-metabolizing enzymes (DME), drug transporters (DTS), and other drug targets [3]. In addition to the presence or absence of genetic variants, developmental differences in protein maturation, co-medications that affect the function of these proteins and disease-related factors also impact results [3]. Both DME and DTS are involved in the absorption, distribution metabolism and excretion (ADME) of the drugs. However, DTS are ubiquitously located throughout the membranes in several organs and are involved in the uptake and efflux of drugs. Thus, variability in DTS function determines drug disposition by affecting the drugs’ movement across organs. Further, they are differentially expressed in different organs. Hence, quantifying the clinical implications of DTS in treatment outcomes is challenging. The prime DTS includes solute carrier (SLC) and ATP-binding cassette (ABC) superfamilies [4].

Several factors, such as age, gender, ethnicity, environmental determinants, organ-related factors, genetic variants, and drug–drug interactions, could influence the DTS gene expression and/or function (Figure 1). For example, the OCT subfamily of SLC22 proteins are highly polymorphic at the genetic level, which influences drug disposition [5]. An interplay between genetic variants and DTS expression has been reported for several diseases. For example, a transient course of neonatal intrahepatic cholestasis is frequent among carriers of predicted biallelic pathogenic variants in *ABCB11*, which was resolved upon improved expression of BSEP [6]. Gender-specific influences on DTS expression have been demonstrated for some DTS. For example, SCL22A12 showed a higher expression in young females (<50 years) compared to young males (<50 years). Further, within females, SLC22A2, SLC22A12, SLC6A16, and ABCB6 were more highly expressed among younger than older individuals [7]. A similar gender-based differential expression of DTS has been demonstrated in the human liver [8]. Ethnicity may affect DTS expression and or function, driven by the genetic variants and their linkage with disequilibrium patterns specific to each ethnicity. ATP7B and KCNJ8 were highly expressed in African-American women compared to European-American women. The expression of SLC31A2 was increased among European American males in contrast to European American females [7]. Drug-induced regulation of gene expression is documented for several drugs with microarray experiments in cell lines [9]. Mithramycin has been shown to alter the expression of various bile transporters in hepatic cell lines, leading to hepatotoxicity [10]. Among the environmental factors, an interaction has been reported between diet, gut microbiota and nutrient transporter expression [11]. Among the diet-related factors, isoflavones, together with the photoperiod duration, influence the gut microbiota and genes such as glucose transporter type 4 (*Glut-4*) and lipoprotein lipase (*Lpl*) [12].

Among all these factors, the determinants characteristic specific to pediatric chemotherapy are DTS expression on tumors and their ontogeny. For example, the up-regulation of inositol-requiring enzyme 1α (IRE1α) and X-box binding protein 1 (XBP1) pathway is shown to cause drug resistance to 5-flurouracil through the expression of efflux transporters, ABCB1, ABCC1, and ABCG2 in colon cancer cells [13]. Poor prognosis due to high levels of MRP1 expression in neuroblastoma with N-myc amplification is also a well-known example of drug resistance due to DTS expression in tumors in children [14,15]. MRP1 is also over-expressed in other pediatric cancers, such as AML and ALL. The MRP1 inhibitor is postulated to reverse drug resistance in tumors overexpressing MRP1 [16]. Another example is the upregulation of MDR1 and BCRP mRNA expression after standard chemotherapy with cisplatin and doxorubicin in hepatoblastoma, the most common malignancy of the liver in children [17]. In children aged from 1 week to 11 years with high-grade glioma, tumor-tissue staining indicated the presence of P-gp, MRP1, and BCRP1 on the tumor vasculature, while MRP1 is also expressed in glioma cells [18]. Thus, DTS in tumors contributes to the chemotherapy drug resistance.

However, DTS at the ADME level leads to changes in drug disposition and, hence, is more likely to show variation due the associated factors (Figure 1). For example, the polymorphisms of ABCB1 resulting in reduced activity at the blood–brain barrier is shown to be responsible for a higher risk of morphine-induced respiratory depression in neonates compared to adults [19]. The role of DTS at the ADME level in the use of chemotherapeutic drugs among the adult population is sufficiently elucidated [20,21]. However, in children, due to their growth and development, another determinant, ontogenesis, play a role. For example, immature OCT1 activity may be associated with lower tropisetron uptake in the liver and its clearance in the first year of life, as demonstrated by the dysfunctional variants in *SLC22A1* that encode the transporter [22]. In addition, DTS could interact with DMEs that also occur in ontogenesis in children, and thus influence drug ADME. Thus, there is a need to understand the role of the developmental patterns of drug transporters in explaining the variability in pharmacokinetics and pharmacodynamics for drugs (chemotherapy) used in the pediatric population [23,24].

In this context, the pediatric transporter working group has given several recommendations regarding the existing knowledge, including the identification of selective or specific biomarkers for the evaluation of transporter activity in pediatric patients, as well as the investigation of basic developmental mechanisms that regulate transporter expression and activity in different organs in pediatric health and disease [23]. Recent advances in modeling, such as physiologically based pharmacokinetic (PBPK) modeling, involve drug- and patient-specific factors, which can also include drug transporter ontogeny information, if available, and can adequately predict pediatric-specific pharmacokinetics of chemotherapeutic drugs or the supportive care therapy used to prevent chemotherapy-related complications [24]. In this narrative review, we emphasize the role of drug transporters (DTS) at the ADME level and the development patterns that are relevant in the pediatric oncology setting.

## 2. Ontogeny in DTS and Possible Consequences for Pediatric Chemotherapy

Developmental changes have been reported for some human transporters, but information and data for many transporters are either limited or unavailable (Figure 2). The recent application of LC-MS/MS in proteomics has facilitated the identification of ontogeny with respect to several transporters with greater precision than in studies of mRNA expression [25]. Ontogeny data are mainly obtained using postmortem and surgical tissue samples by implementing various techniques, such as immunohistochemistry to visualize tissue localization, reverse transcriptase polymerase chain reaction or microarrays for messenger RNA expression, Western blotting and LC-MS/MS techniques to quantify the abundance of transporter proteins [23]. The availability of quality paediatric tissue is the major hurdle in drug transporter research in the pediatric population. Animal data on the developmental patterns of membrane transporters may not be completely reliable, as there will be differences with regard to the development of these transporters between humans and animals [26]. Further, within human studies, transporter development patterns to date have highlighted the lack of a correlation between mRNA and protein expression [27]. The interpretation of ontogeny data also depends on the methodology used.

The protein expression data (Figure 2) are not in complete concordance with mRNA expression data (Figure 3 and Figure 4) [28,29], indicating the regulation of DTS at various levels, and stressing the need to implement a method of monitoring DTS function in real-time. The DTS function in real-time can be assessed using probe drugs and by measuring their levels in vivo at nontoxic concentrations. In vivo probes for individual drug-metabolizing enzymes (DMEs) have previously been reviewed [30]. Such in vivo probes have been proposed for DTS such as P-gp, BCRP, OATP1B1, OATP1B3, OCT2, OAT1 and OAT3; however, the specificity of these probes is still uncertain, as multiple transporters are involved in their disposition [31,32]. In addition, validation of these probes in the pediatric population is essential, due to the altered dosing strategies and the safety of these agents.

Among the many human membrane transporters, data on ABCB1 are comparatively higher. The ABCB1 developmental pattern is organ-specific. In the intestine, stable expression of ABCB1 is seen from neonatal to adult life, as evidenced by intestinal mRNA data [35]. However, in the ABCB1 liver, expression is low in fetuses, neonates, and infants up to 12 months of age, before subsequently increasing to adult levels [36]. Intestinal P-gp mRNA expression is shown to exhibit high inter-individual variations [35,37]. The available data suggest that the intestinal P-gp reaches adult levels at or shortly after birth [38]. Children treated for Crohn’s and celiac disease showed higher intestinal P-gp expression compared to healthy controls [39]. Similarly, intestinal MRP2 mRNA expression is stable from neonates to adults [36]. However, OATP2B1 mRNA expression seems to be higher in neonatal samples in contrast to adult samples [36]. Other intestinal transporters, such as BCRP, are stable from 5.5 to 28 weeks of gestation in humans; MRP1 matures to an adult distribution pattern from 7 weeks of gestation onwards [40].

In the liver, some transport proteins, such as BCRP, MATE1, MCT1, and OATP2B1, are completely expressed from early life to adulthood, with stable expression throughout [25,33]. P-gp, MRP2, MRP3, and OCT1 show a lower expression in early life compared to adulthood [33], while transporters such as GLUT1 have been shown to have a higher expression during the fetal stages compared to adult tissues [41]. Data on the expression of BSEP, MRP1, NTCP, OATP1B1, and OATP1B3 vary between studies [41]. However, a recent study, which included a high number of fetal and neonatal samples, suggested that BSEP, MRP1, and NTCP show lower expression in early life and gradually match adult levels. OATP1B1, however, is shown to have higher expression during fetal stages than at adult levels [41]. OATP1B3 has been shown to be stable throughout age groups [41].

With regard to transporters in the kidney, animal studies indicate the low expression of various transporters in the late stages of kidney development, followed by a rapid rise in expression after birth, which further increases during postnatal maturation. The expression in the kidney is detectable as early as 11 weeks of gestation [23]. A recent study involving larger samples from neonates and infants has shown P-gp, OAT1, OAT3, and URAT1 to be less expressed in early life compared to adults [34]. The data vary for OCT2 [34,42]; however, it has a lower expression in early life [34] (Figure 2).

The ontogeny of DTS in the blood–brain barrier has not been well-studied owing to the scarcity of post-mortem tissue sample collections in pediatrics. However, DTS in the BBB plays a key role in pediatric oncology. The documented role of ABCB1 in busulfan disposition is relevant for adverse drug reactions, such as neurotoxicity during hemopoetic stem cell transplantation [43,44]. Nevertheless, age is shown to be a key determinant in the quantitative proteomics of ABCB1 in post-mortem adult human microvascular endothelial cells [45]. ABCB1 was also shown to have a reduced expression in men compared to women. Furthermore, ABCB1 was more highly expressed in the occipital lobe compared to parietal lobe [45]. The use of positron emission tomography using carbon 11-labelled verapamil as an ABCB1 substrate in humans has been validated [46,47]. Functional phenotyping studies may pave the way to studying DTS ontogeny in BBB in the context of pediatric oncology. Thus, the differential maturation of key DTS across age groups should be considered while evaluating variability in treatment outcomes. However, data on the DTS involved in a specific drug, PK or PD, are largely derived from adults.

## 3. Role of DTS in Chemotherapy-Related Toxicities and Response in Pediatric Oncology

In this section, we present a brief overview of the available evidence regarding the role of DTS in determining the therapeutic response and toxicities to frequently used chemotherapeutic agents in pediatric oncology (Table 1). The drugs listed in the National Cancer Institute list of drugs, approved by the United States Food and Drug Administration and commonly used for childhood cancers, were selected [48]. The review does not include drugs commonly used in the hematopoietic stem cell transplantation (HSCT) or chimeric antigen receptor (CAR)-T cell therapy (e.g., busulfan, treosulfan, fludarabine) settings, or drugs used off-label in the pediatric population (e.g., cisplatin for osteosarcoma) [49]. The narrative description is based on the following assumptions: (a) variants in DTS-encoding genes might explain the variability in toxicities indicating the functional role of DTS; (b) developmental changes may also reflect the scenario of nonfunctional DTS in certain patient age groups, or the impact of immature DTS may be more apparent in reduced function allele carriers in genes encoding DTS; (c) competition among several substrates or interaction with DTS inhibitors may be more apparent in certain age groups, reflecting developmental changes, especially in cancer chemotherapy with multiple substrates or co-medication, such as anti-infective therapy (e.g., azole antifungals); (d) the function and expression of DTS in normal tissues is usually considered as a contributor to the inter-individual variability in treatment (e) all these factors may contribute to the differences in the outcomes that may be seen in different age groups, modulated, in turn, by the genetic variants. Below, we provide an elaborated account of the role of DTS in treatment-related effects for individual drugs that are approved for use in pediatric oncology.

## 4. Enzymes

### Asparaginase (Asparaginase Erwinia Chrysanthemi, Calaspargase Pegol-mknl, and Pegaspargase)

L-asparaginase is a principal component of the chemotherapy regimen for the treatment of acute lymphoblastic leukemia (ALL) in children and adolescents. It acts by depleting asparagine in malignant cells that are incapable of de novo synthesis of asparagine. Besides hypersensitivity reactions, the drug can precipitate pancreatitis and thrombotic events (Table 1) [61]. In vitro studies have shown that the increased expression of the high-affinity glutamate transporter family and neutral amino acid transporters, encoded by genes *SLC1A3-5*, can contribute to the development of resistance when completely developed by improving the supply of amino acids to tumor cells, provided they do not carry dysfunctional variants [111,112]. Although the direct involvement of drug transporters in the definition of the action or toxicity of asparaginase is not known, there is some evidence that the efflux transporter MDR1 or P-gp (encoded by *ABCB1*) may contribute to the development of resistance [113,114]. An increased expression of *ABCC* 2 to six was observed in relapsed pediatric ALL patients compared to non-relapsed patients, who also showed a decreased expression of *ABCC1.* It is also interesting to note that *ABCC* 4 and 6 was higher among those who received high doses of daunorubicin and L-asparaginase [60].

## 5. Purine Nucleoside Analogues

### 5.1. Clofarabine

Clofarabine is indicated for the treatment of refractory or relapsed ALL in patients between 1 and 21 years of age. It is also widely used in the treatment of acute myeloid leukemias (AML) and in hematopoietic stem cell transplantation (HSCT) settings. Important adverse effects include bone marrow suppression and systemic inflammatory response syndrome (Table 1) [35]. The drug is a substrate for sodium-dependent concentrative nucleoside transporters (CNs) encoded by the *SLC28* family, and equilibrative nucleoside transporters (ENTs), a facilitated diffusion carrier encoded by the *SLC29* family. These transporters are expressed at epithelial cell apical surfaces and distributed across various tissues, and may mediate the cellular uptake of clofarabine, along with other nucleoside analogs [115]. However, the expression of ENT transporters in blast cells and the PBMCs of pediatric patients with refractory/relapsed ALL (a low number of patients) was not related to treatment response [116,117]. OAT1, OAT3, and OCT1 are also suggested to be involved in clofarabine uptake in specific tissues. These transporters may mediate clofarabine-induced cardiotoxicity, hepatotoxicity, and nephrotoxicity, although there is no direct clinical evidence available at present [118,119].

Clofarabine kidney excretion has been shown to be decreased by cimetidine, an inhibitor of hOCT2, necessitating caution when this is used in combination with substrates or inhibitors of OCT2 [61]. *ABCG2* (BCRP), *ABCC4*, and *ABCC5* are known to be involved in the efflux of clofarabine but not its intracellular phosphorylated forms. Reducing deoxycytidine kinase levels has been shown to improve ABCG2-mediated resistance to clofarabine [99,100]. The distribution of these transporters in the renal tubule mediates both the secretion and reabsorption of clofarabine [95,120]. Ethnic variations are observed in non-synonymous variants affecting protein function; for example, 35% of Asians and <1% of African Americans carry non-synonymous *ABCG2* variants [121]. Increased clofarabine exposure in adults compared to children indicates the variations in the developmental differences (ontogeny) in transporters involved in the distribution of clofarabine [122].

### 5.2. 6-Mercaptopurine (6-MP)/Azathioprine

6-MP is widely used as a component of the maintenance treatment of various pediatric ALL treatment regimens. *ABCC4* polymorphisms have been shown to be associated with thiopurine metabolism (6 TGN/6 MP dose ratio) [123]. *ABCC4*, *SLC19A1*, *SLC28A3*, *SLC29A1*, *SLCO1B1* genetic variants were associated with thiopurine-related toxicities [123,124]; neutropenia [125,126], hepatotoxicity and treatment interruption [123], especially in Asians. *NUDT15* and *ABCC4* are the main factors in the intolerability of 6-MP the interaction between these variants enhances intolerability with 6-MP [127].

Evidence from studies in ALL children indicates significant associations between *SLC29A1*, *SLC28A2*, *SLC28A3* and *ABCC4* variants and one or more of the hematological toxicity manifestations (neutropenia, agranulocytosis and leukopenia) [128]. *SLC29A1* encodes ENT1, which mediates the uptake of several nucleosides; *SLC29A1* expression was high in ALL cells that accumulated high post-treatment thioguanine concentrations [129]. Polymorphisms in *ABCC5* have also been shown to explain the clinical response and levels of thiopurine [101]. Both *ABCC4* and *ABCC5* participate in the efflux of mercaptopurine metabolites [101]. The role of the *ABCB1* 1236C>T variant in the requirement for mercaptopurine dose and the incidence of hematologic toxicity should not be overlooked [102].

In vitro evidence shows that the up-regulation of MRP4 and down-regulation of influx transporters hENT1, hCNT2 and hCNT3 play a major role in the 6-MP resistance of 6-MP-resistant human T-lymphoblastic leukemia cell line [130]. Thus, the available literature indicates the role of multiple influx and efflux transporters in defining the response to treatment and toxicities of 6-MP.

### 5.3. Nelarabine

Nelarabine is a water-soluble pro-drug of arabinosylguanine (ara-G), a nucleoside metabolic inhibitor, indicated for the treatment of patients with T-cell ALL and T-cell lymphoblastic lymphoma [61]. The ara-G enters leukemic cells via ENT1 and ENT2 [104]. Evidence from bone marrow/peripheral blood samples from 96 pediatric acute leukemia patients untreated suggested that ara-G resistance could be mediated by a lower expression of ENT1 and ENT2 [103]. An in vitro study found that *SLC29A1* gene expression levels were highly associated with ara-G sensitivity in human T-ALL cell lines [104]. Neurotoxicity, the main adverse effect of nelarabine, requires the uptake of the drug into the neuronal cells; however, ENT’s tole in elucidating this toxicity is unknown [131].

## 6. Pyrimidine Nucleoside Analogues

### Cytarabine

Cytarabine is indicated in ALL and chronic myeloid leukemia (CML); bone marrow suppression is the most prominent adverse effect [61]. *ABCC4* is abundantly expressed in myeloid cells, where it facilitates the efflux of cytarabine and its monophosphorylated metabolite. ABCC4-mediated efflux plays a protective role against cytarabine-mediated insults in leukemic and host myeloid cells, thereby decreasing myelotoxicity [109]. Among the other members of the ABCC family, the expression of ABCC3 (MRP3) was shown to influence treatment in childhood AML, although the study did not demonstrate direct transport of the cytarabine metabolite [106]. Cytarabine is also a substrate of ABCC10 (MRP7) [79] and ABCC11 (MRP8) [132]. Further, high ABCC11 expression has been associated with a low probability of overall survival (OS) in AML adults, and should be evaluated in children of different age groups [132].

The ABCB1 TTT haplotype was related to OS, relapse-free survival (RFS) in Chinese patients with AML who received anthracycline and cytarabine [107]. However, a meta-analysis did not find conclusive results linking ABCB1 expression to OS in AML [108]. A study among 282 pediatric cases of AML that assessed event-free survival (EFS) and the OS effects of standard and high-dose induction with or without cyclosporine found that no significant survival benefit was observed; overexpression was rare (14%) in this pediatric population [133].

Similar to other nucleoside derivatives, cytarabine enters the cells via ENT1 (*SLC29A1*) [105], and low uptake predicts poor response [134]. FLT3 tyrosine-kinase receptor expression regulates hENT1 activity and, hence, the sequence of administration of cytarabine and FLT3 inhibitors, such as sorafenib, is important. The cytotoxicity of cytarabine is lower when cells are first exposed to FLT3 inhibitors, probably due to the decreased hENT1 activity [135]. An ENT1-independent pathway occurs via OCTN1 (*SLC22A4*; ETT). Its low expression in leukemic cells is a strong predictor of poor survival in multiple cohorts of patients with AML treated with cytarabine-based regimens [109]. An evaluation of 103 patients with AML showed that reduced expression of *SLC29A1* was significantly associated with reduced cytotoxicity [136]. *SLC28A3* (CNT3) may also be involved in cytarabine transport [106]. Given the role of transporters, one may assume their role in the toxicities can be seen with cytarabine treatment, especially when combined with alkylating agents.

## 7. Folate Antagonist

### Methotrexate

Bone marrow suppression, hepatic and renal injury due to methotrexate (MTX) are dose-limiting adverse effects in ALL patients. High variability has been found in intracellular methotrexate (MTX) polyglutamate forms, which influences the drug response. ALL children with a relatively lower expression of *SLC19A1* compared to *ABCC1* and *ABCC4* in leukemic cells exhibited a poor response to high-dose MTX [75]. These transporters determine the drug’s accumulation in the cancer cells. ABCC4 and ABCG2, present in leukemic cells, efflux both MTX and its polyglutamate forms and decrease the response in childhood ALL [137].

SLCO1A2 cellular uptake transporter, rs4149009 G>A polymorphism, is associated with delayed MTX elimination [72]. However, several cellular influx transporters located in hepatocytes and renal tubular cells may be responsible for the DDIs seen with MTX. For example, dasatinib inhibits the transporter SLCO1B1 and causes a delay in MTX clearance [73]. A study in Turkish ALL children showed that carriers of variant alleles in SLCO1B1 rs4149056 and rs11045879 had a lower tolerance [124]. SLCO1B1 is a sinusoidal transporter that mediates the liver’s elimination of MTX; it is an important determinant of methotrexate pharmacokinetics and clinical effects [138]. The most clinically concerning adverse reaction to high-dose MTX is acute kidney injury, with a high incidence of 2–12% and a mortality rate of 4.4%. The SLC22A6 (OAT1) and SLC22A8 (OAT3), present on the basolateral membrane, and ABCC2 (MRP2) and ABCG2 (BCRP), present on the apical membrane, are responsible for the clearance of MTX. Inhibitors of these transporters or dysfunctional variants in these transporter genes can delay MTX clearance and predispose one to toxicity [57]. The drug interaction involving hOAT1 (SLC22A6) and hOAT3 (SLC22A8), responsible for renal tubular methotrexate secretion, has also been described in a Japanese patient with Hodgkin’s disease [70].

Free MTX is exported from the cell through ABCC1–5, ABCC10, and ABCG2 transporters, and a number of studies have found that SNPs in genes coding these transporters are associated with MTX toxicity [71,102,139,140,141,142]. Further, a higher concentration of MTX polyglutamates in plasma has been attributed to ABC transporters [143].

## 8. Microtubule Inhibitors

### Vincristine

Vincristine undergoes biliary excretion via P-glycoprotein, which is expressed at approximately 60% and 80% of adult levels in infants and children, respectively [144]; however, a study among 26 children treated for various solid tumors suggested that P-gp is not a major factor in drug clearance [81]. However, polymorphisms in the ABCB1 modulating vincristine neurotoxicity have been observed in childhood ALL patients [82].

A retrospective evaluation of 133 Arab children with ALL did not show a relationship between *ABCB1*, *ABCC2* genetic variants and vincristine-induced peripheral neuropathy [83]. However, another study found that *ABCC2* genotypes rs3740066 GG and rs12826 GG were associated with increased neurotoxicity [145]; an association with *ABCC1* rs3784867 [146] and *SLC5A7* rs1013940 was also seen, with the latter, encoding a choline transporter, being involved in inherited neuropathies.

The high expression of the *ABCC1* and *RALBP1* genes [78] could result in an elevated vincristine efflux from the cell, which explains the decreased risk of drug-induced neurotoxicity. Even though the effect of rs12402181 seems to decrease cell sensitivity to vincristine, when OS and EFS were analyzed, no association was found [147].

*ABCC3* SNP rs4148416 was associated with an 8-fold risk of low survival in 102 osteosarcoma patients (aged 3–34 years) who received vincristine in conjunction with other drugs [80]. An in vitro study has also shown that ABCC10 has a possible role in mediating vincristine resistance [79].

## 9. Alkylating Agents

### Cyclophosphamide

Cyclophosphamide is another anti-ALL prodrug that can lead to decreased blood cell counts and hemorrhagic cystitis. Most studies of drug transporters in relation to cyclophosphamide have occurred in patients on combination regimens. Data from a study in breast cancer patients show that ABCC4 expression is associated with gastrointestinal toxicity and leukopenia. In particular, ABCC4 in the kidney is responsible for the excretion of the drug and its metabolite; hence, its underexpression contributes to systemic toxicity. In contrast, in the GIT, the transporter is responsible for the efflux of the drug into the systemic circulation, thereby contributing to its toxicity [148]. However, the expression of the transporter and its effect on drug kinetics and dynamics vary depending on the disease condition. No link to toxicity was observed for ABCC4 in adult and pediatric patients who received an allogeneic blood and marrow transplant [50]. ABCC2 was shown to be involved in the clearance of cyclophosphamide in patients > 18 years of age who underwent autologous stem cell transplantation [51]. ABCB1 is associated with an increased risk of death or adverse drug reactions in breast cancer patients who receive adjuvant chemotherapy with cyclophosphamide and doxorubicin [52]. The inhibition of ABCB1 was associated with vincristine, actinomycin-D, and cyclophosphamide regimen toxicity in osteosarcoma [149]. However, despite its overexpression, ABCB1 was not a significant factor determining early response in childhood ALL [113], again highlighting the differential effect of transporter expression based on disease and other factors. Other transporters that may be associated with cyclophosphamide handling are ABCC3, ABCG2, SLC22A16, and SLCO1B1 [52,150].

## 10. Topoisomerase Inhibitors & Transcription Inhibitor

### Doxorubicin, Daunorubicin (Anthracyclines), and Actinomycin-D

Doxorubicin and daunorubicin are important cytotoxic drugs for childhood cancers. Cardiotoxicity is an important adverse effect, which is directly caused by doxorubicin and, in particular, by the daunorubicin metabolite [151]. In a genetic association study of 196 patients, including children with high-grade osteosarcoma receiving methotrexate, doxorubicin, cisplatin and ifosfamide carriers of the *ABCC2* variant allele c.1249A>G were associated with worse EFS and grade 4 leukopenia, with an increased need for red blood cells and platelet transfusion [56]. *ABCC2* polymorphism (rs8187710, 1515G>A), among other factors, has been shown to be associated with increased odds of congestive heart failure (CHF) in patients, including children, at risk of CHF after hematopoietic stem cell transplantation with doxorubicin received pre-HCT [54]. In childhood ALL, *ABCC5* TT-1629 genotype demonstrated a cardiotoxicity protective feature in 8–12% of patients [57]. The *ABCB1* 1199G>A variant was related to the risk of relapse and 3435TT genotype was associated with bone marrow toxicity in ALL [55,152].

Several transporters have been shown to modulate daunorubicin pharmacokinetics. In a study of Iranian pediatric ALL patients, a high expression of ABCC2–6 and low expression of ABCC1 were detected in patients with relapse. ABCC4 and ABCC6 expression increased with high doses of daunorubicin and L-asparaginase [60]. Daunorubicin is one of the substrates of OATP1B1 (SLCO1B1). A study of 164 pediatric AML patients showed that a synonymous SNP rs2291075 was significantly associated with EFS and OS, with carriers not having worse survival. OATP1B1 mediates the hepatic uptake of drugs [109]. The expression of OATP1B1 was found to be significantly lower in all pediatric age groups compared to adults [23]. Both doxorubicin and actinomycin-D are components of Wilms tumor chemotherapy; a high expression of MDR3 and MRP1 has been shown to confer resistance in children with Wilms tumor and a significant correlation has been demonstrated with clinicopathological parameters [58]. A case report linked *ABCB1* gene polymorphism to actinomycin-D-induced hepatopathy-thrombocytopenia syndrome [63]. However, a clinical pharmacokinetic study of 117 children with cancer found no significant relationships between *ABCB1* genetic variants and actinomycin-D pharmacokinetic parameters or incidence of grade 3/4 toxicity (Table 1) [62].

Similar discordant findings are also seen in studies in adult patients, with some linking transporter function to risk of treatment failure [153] while others fail to demonstrate such a relation [154]. The role of transporters in the placenta has also been explored in few studies. P-gp and ABCG2 (BCRP) are expressed in the placenta. P-gp expression in the syncytiotrophoblast microvilli that border the maternal side of the human placenta decreases from the 14th to 40th weeks of gestation, suggesting a gradual decrease in fetal efficacy and protection. Although its clinical consequences are still little known, the decreased expression/function of P-gp in the placenta might increase the distribution of the drug substrate to the fetus, possibly intensifying its effects [155].

## 11. Kinase Inhibitors

### 11.1. Imatinib

Indicated for the treatment of Philadelphia chromosome-positive chronic myeloid leukemia, imatinib can precipitate cytopenias and heart failure, in addition to other serious adverse drug reactions [61]. Imatinib showed some inhibitory effect on SLC6A4, which is responsible for serotonin reuptake; inhibition results in increased plasma serotonin levels, leading to diarrhea. However, the study did not directly evaluate the inhibition of the transporter by the drug, and the transporter inhibition only explains a small percentage of the cases [156]. There is little evidence linking transporter function to drug toxicity. Most studies demonstrate the effect of various transporters on imatinib pharmacokinetics/dynamics, which may be inferred from the chance of adverse drug reactions.

*SLC22A1*, which encodes the OCT-1 uptake transporter, is an important determinant of imatinib concentration within leukemic cells and also affects the liver clearance of the drug [86,87]. The *hOCT1* gene was significantly down-regulated in patient samples with resistance to imatinib compared with the responder group [157]. The intracellular imatinib concentration was significantly higher in patients with *SLCO1B3* 334TT than in those with 334TG/GG [158]. In contrast, a Japanese study found that individual clearance significantly increased in those with the *SLCO1B3* 334GG genotype compared to the *SLCO1B3* 334TT and TG genotypes [159]. A microarray-based analysis studying glycolysis/glycogenesis-related gene expression in childhood B-cell ALL showed that imatinib can repress SLC2A5 expression and decrease fructose uptake in cells, contributing to cytotoxicity [160].

ABCB1 plays an important role in determining the response to imatinib. An early evaluation of the fold change in *ABCB1* mRNA expression may identify patients that are likely to be resistant to first- and second-generation TKIs and who may be candidates for alternative therapy [89]. A population PK study of imatinib in Nigerians showed that *ABCB1* C3435T polymorphism affected the drug clearance, with the highest clearance being seen in those with the CT polymorphism [161]. In a Japanese population pharmacokinetic study, patients with the ABCB1 3435CC genotype showed significantly higher clearance of imatinib compared to patients with the ABCB1 3435CT and TT genotypes [159]. Although the role of ABCB1 in improving drug clearance and decreasing toxicity is clear, there is no consensus on the exact polymorphism responsible for this [162,163,164]. Moreover, not all studies support ABCB1’s effects on imatinib clearance. White et al. and Yamakawa et al. found no correlation between imatinib concentrations and SLCO1B1, ABCB1, and ABCG2 [86,159]. Singh et al. showed a functional dependency of ABCB1 on SLC22A1 [87], which may explain the negative findings regarding the efflux transporter in some studies. Additionally, many of the polymorphisms observed in Caucasian studies were not seen in Asian patients, as well as the interethnic differences in these polymorphisms [87]. The meta-analysis by Zheng et al. showed that the 2677G or 3435T allele predicted a worse response to imatinib in CML patients, whereas the 1236CC genotype was associated with better response in CML patients from Asian regions [88]. Although ABCB2 has also been suggested to alter imatinib kinetics [165], this has not been adequately clinically proven [166,167]. It should be noted that all the above observations were from studies conducted in adult patients. Hence, a study of genetic polymorphisms in the pediatric population is required.

### 11.2. Nilotinib

Nilotinib is approved for use in children > 1 year of age for newly diagnosed chronic-phase CML, including those who cannot be treated or who did not respond to tyrosine kinase inhibitor therapy [168]. However, no studies have evaluated the role of transporters in drug toxicity or response in the pediatric population. In vitro studies have shown that nilotinib efflux may be mediated by ABCB1 and ABCG2, conferring resistance, although the relative importance of each of these transporters has not been established [84]. Whether ABCB1 mediates the transport of nilotinib may also be dependent on the variant that is expressed [90,152]. Nilotinib has been suggested to have an inhibitory effect on transporters [169]; a study of bone marrow samples of 87 patients with CML at diagnosis showed decreased expression of ABCC4, ABCC5, and ABCD3 at the end of 12 months [170]. Conversely, in nilotinib-resistant cells, an increased expression of ABCC6 has been demonstrated [91].

### 11.3. Dasatinib

Dasatinib is a second-generation, tyrosine kinase inhibitor approved for Philadelphia chromosome-positive CML and ALL in patients who are resistant or intolerant to imatinib. It is a substrate for the ABCB1 and ABCG2 [84]. Dasatinib is also an inhibitor of ABCB1; however, this occurs at a much higher concentration than is clinically attained in CML patients [85].

### 11.4. Crizotinib

Crizotinib is approved for use in children 1 year of age and older and young adults with systemic anaplastic large-cell lymphoma (ALCL), positive for ALK, relapsed, or refractory. Adverse effects include ocular toxicity, hepatotoxicity, and interstitial lung disease [61]. No data are available on DTS in the pediatric population. Crizotinib is a substrate of P-glycoprotein [171]. In a small study among Japanese adult patients with non-small cell lung cancer, a patient with an *ABCB1* genotype (all TT) had elevated concentrations of the drug and developed grade 3 prolongation of QT, suggesting an increased susceptibility to adverse effects [93]. Similarly to other tyrosine kinase inhibitors, crizotinib has been shown to inhibit ABCB1, OCT3, OCT1/2, OATPB1/3, OATP2B1 and OATP4C1 in in vitro studies [92,172,173]. Entecavir can inhibit crizotinib transport by OCT2 in kidney and increase the risk of vision disorders, diarrhea, and vomiting by 1.6-, 2.3-, and 1.8-fold, respectively [174]. By inhibiting OCT2 and MATE1, crizotinib and imatinib can increase serum creatinine concentration by more than 10% [175,176]. OATPB1/3 and OATP2B1 may mediate hepatic uptake of crizotinib and contribute to drug–drug interactions and hepatotoxicity [177].

### 11.5. Entrectinib

Entrectinib is a tyrosine kinase inhibitor approved for use in pediatric patients 12 years and older with solid tumors and fusion of a neurotrophic tyrosine receptor kinase (*NTRK*) gene. In vitro studies show that the drug is not a substrate for P-gp or BCRP, but its metabolite M5, can be used in this way [61]. A potential inhibitory effect has been described on P-gp [94], BCRP, OATP1B1, and MATE1, although clinical significance has not been established [173].

### 11.6. Larotrectinib

Larotrectinib is another kinase inhibitor approved for use in pediatric solid tumors that has a *NTRK* gene fusion without a known acquired resistance mutation [61]. No drug transporters have been implicated in the causation of neurotoxicity. In vitro studies have shown that it is a substrate of ABCB1 and ABCG2, which limits its oral availability, as well as its penetration into the brain [61,95]. OATP1A/1B transporters restrict its systemic exposure by mediating hepatic uptake, thus allowing for hepatobiliary excretion [95]. Unlike entrectinib, it is not an inhibitor of any of these transporters [61].

### 11.7. Selumetinib

Selumetinib is a mitogen/extracellular signal-regulated kinase inhibitor indicated for the treatment of pediatric patients 2 years of age and older with neurofibromatosis type 1 (NF1) who have symptomatic, inoperable plexiform neurofibromas [61]. Skin and gastrointestinal toxicities are common, but cardiac and ocular toxicities can also occur. ABCB1 and ABCG2 efficiently limit its penetration through the blood–brain barrier [97]. Dymond et al. conducted a pooled analysis of phase 1, single-dose studies of selumetinib in healthy subjects of Asian and western ethnicity; the study showed that drug exposure was higher in Asian subjects compared to western subjects, but there was no evidence that the polymorphisms in *ABCG2* or drug-metabolizing enzyme genes accounted for the observed pharmacokinetic differences [96].

### 11.8. Everolimus

Everolimus is an inhibitor of mammalian target of rapamycin (mTOR), a serine-threonine kinase; it is indicated for use in children with subependymal giant cell astrocytoma. It is a substrate for P-gp and requires a dose reduction when administered concomitantly with P-gp inhibitors [61,178]. In a pharmacogenetic analysis of blood samples from 90 postmenopausal women with hormone receptor-positive, HER2-negative metastatic breast cancer treated with exemestane-everolimus, *ABCB1* rs1045642 was associated with a significantly increased risk of mucositis [98]. Everolimus has been shown to inhibit OATP1B1, OATP1B3, OATP1A2 [179].

## 12. CD33-Directed Antibody-Drug Conjugate

### Gemtuzumab Ozogamycin

Gemtuzumab ozogamicin (GO) is an immunoconjugate between an anti-CD33 antibody and N-acetyl g1 calicheamicin. The latter is responsible for the cytotoxic action after lysosomal enzymes cleave the conjugate drug, leaving calicheamicin, which forms a highly reactive radical. The antibody serves to target the cytotoxic conjugate to the CD33-expressing AML blast cells [180]. GO is indicated in the treatment of patients with AML 1 year and older, with its use being associated with better EFS [181,182,183]. In a genetic study of 94 CD33-positive Italian AML patients younger than 65 years receiving a combination of low-dose GO with Fludarabine, Cytarabine, and Idarubicin as an induction chemotherapy, the *SLCO1B1* rs4149056-derived allele responsible for the uptake of drugs from blood to hepatocytes was overrepresented in patients with liver toxicity. Further, rs11231825 that was located on the *SLC22A12* gene encoding the urate transporter has been shown to be associated with fever reaction following GO infusion [184].

The ABCB1-mediated efflux of calicheamicin might decrease the clinical utility of the drug. In a study of cell samples obtained from children and adults (0–29 years) with AML, randomized to chemotherapy regimens with or without GO, the low-expressing *ABCB1* rs1045642 TT genotype increased intracellular calicheamicin levels in leukemic cells due to a reduced efflux, enhancing chemosensitivity to GO [69]. The role of ABCB1 in influencing the GO cytotoxicity of GO has also been demonstrated in adult patients with AML [185,186]. An interplay between CD33 and ABCB1 expression has also been demonstrated in children; in adults, no differences could be demonstrated, despite a similar genotype distribution; the abundance of ABCB1 in adults has been proposed as the possible reason for this [187]. Boyer et al. suggest that ABCB1 has a prognostic role in patients with AML, but may not affect the treatment outcome [68]. A possible role of ABCC1 (MRP1) in resistance to GO has also been suggested (see Table 1) [67]. Some of the drugs approved for use in pediatric cancers, such as the immune checkpoint inhibitors ipilimumab, nivolumab, pembrolizumab, and avelumab; blinatumomab and tisagenlecleucel; GD2-binding monoclonal antibodies dinutuximab and naxitamab, do not have evidence for the involvement of any drug transporters, thus influencing their adverse effects or response.

## 13. Future Perspectives and Conclusions

The era of personalized medicine in pediatric oncology has reduced mortality and morbidity to some extent. However, there is a large scope for the further optimization of therapy in this population. The evolving nature of DTS in children plays an important role in the pharmacokinetics and pharmacodynamics of oncotherapeutics. Thus, understanding the ontogeny pattern of DTS is a crucial step toward therapy optimization and preventing drug–drug interactions. For example, clofarabine is a drug with significant renal elimination in its unchanged form and OCT2 is involved in its elimination [188]. A population pharmacokinetic study, involving data from 53 children and 91 adults, showed that drug exposure increased with age [100]. The authors proposed that the difference in transporter expression could be a potential basis for this. This is, to some extent, supported by studies on DTS ontogeny (Figure 2). However, further studies or PBPK modelling strategies using the limited available data are required to establish this relation. Similarly, ontogeny studies showed that OCT1 and ABCB1 expression in newborns and infants is significantly lower compared to adults (Figure 2); however, there is no clinical evidence that any adverse effects with chemotherapy can be attributed to transporter expression. MTX is another example of a drug that is significantly affected by DTS, but the only evidence available pertains to genotype and not age-related differences [189]. Given the lack of evidence, it is necessary to consider whether this is due to the relatively smaller effect of age-related transporter expression compared to the established effects of genotype variations and other physiological changes related to age and body size. The available ontogeny evidence shows that transporter expression in children and adolescents, in contrast to neonates and infants, is quite close to adult expression levels (Figure 2).

The availability of tissue samples and variability in methodologies are key challenges to the creation of ontogeny data. Proteomics through LC-MS/MS is a better approach to studying expression patterns. However, in vivo probe drugs would reveal the dynamic function of the DTS proteins. Both these approaches should be used to assimilate data on ontogeny, especially for key DTS, such as p-glycoprotein and SLC. Different substrates were established as probe drugs for DTS activity assessment in vivo. For example, digoxin was used in combination with rosuvastatin to evaluate the activity of the P-glycoprotein, OATP1B1, OATP1B3, and BCRP, respectively [190]. Digoxin and pravastatin were also used as part of a cocktail for the evaluation of P-glycoprotein, OATP1B1, OATP1B3, and MRP2 activities [191]. Endogenous biomarkers represent promising probes for the characterization of drug transporter functions. Bile acids, bilirubin and its conjugates, tryptophan, and taurine were used as markers of OAT, OATP, OCT, MRP2 and MATEs transporter activity [192,193,194,195].

Recently developed drugs are generally well-characterized regarding their interaction with DTS, using different in vitro models. However, no studies are available for older drugs. A comprehensive screening of these drugs’ interaction with different transporters could help to better prescribe drugs and predict drug–drug interactions. Several in vitro and in vivo models were developed to characterize drug transport and drug–drug interactions via DTS. Caco-2 and MDCK cells are widely used to assess the activity of different transporters. Membrane vesicles and cDNA encoded transfected cell lines, expressing one specific transporter, are useful for the rapid screening of the interaction between drugs and transporters and the determination of transport kinetic parameters. Primary human hepatocytes, in 2D or 3D, represent a gold standard in vitro model for the evaluation of metabolism, transport, and the interaction between both mechanisms. Drug–drug interactions involving transporters are also an important issue in drug response variability. In vitro models can help predict the clinical importance of such interactions. Kinetic parameters generated from in vitro experiments (Jmax, Km, Ki, Kinact, IC50, Indmax, IndC50, etc.) could help predict such interactions. They can be implemented in PBPK models for in silico prediction. PBPK modeling is increasingly used in drug development. This approach integrates population- and drug-specific parameters. In the pediatric population, these models integrate the biological maturation processes of metabolic enzymes and transporters in different periods of childhood. Genetic polymorphisms and all other physiological parameters can be used to increase confidence in the predictions. Data on ontogeny also pave the way for accurate PBPK models that, in turn, facilitate clinical drug development in children. Further, well-designed clinical studies are needed to predict and evaluate DTS protein expression and its interplay with other determinants of treatment response or toxicities in pediatric oncology.

Tumoral tissue and cancer cells generally increase the expression of transporters as a protection system, leading to resistance to chemotherapy. The characterization and determination of DTS expression and activity in tumor tissues and cancer cells can considerably improve anticancer therapy by modulating its activity using specific DTS inhibitors or using drugs that are not substrates of overexpressed DTS. However, no data are available on the differences in DTS over-expression in tumoral tissues and cancer cells in children compared to adults. The use of PBPK modeling, coupled with quantitative proteomics or the mRNA expression of different transporters in tumor biopsies or cancer cells, could help to predict drug penetration into the targeted tissue or cells. One must note that DTS function is only one of the factors that need to be considered when optimizing chemotherapy dosing in children.

## Figures and Tables

**Figure 1 pharmaceuticals-15-00990-f001:**
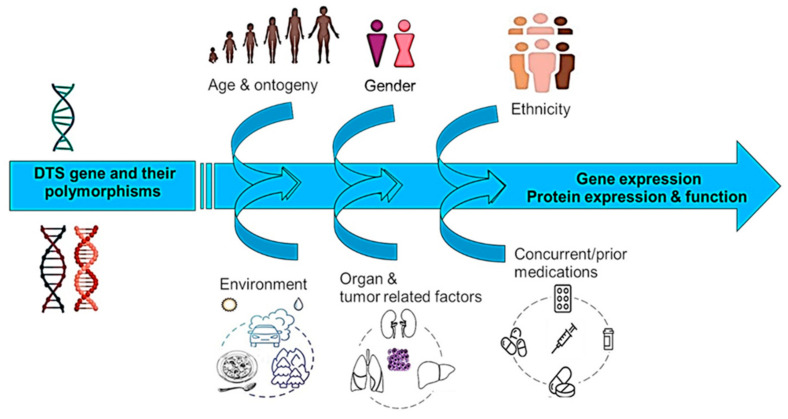
Factors determining expression and function of drug transporters in pediatric chemotherapy.

**Figure 2 pharmaceuticals-15-00990-f002:**
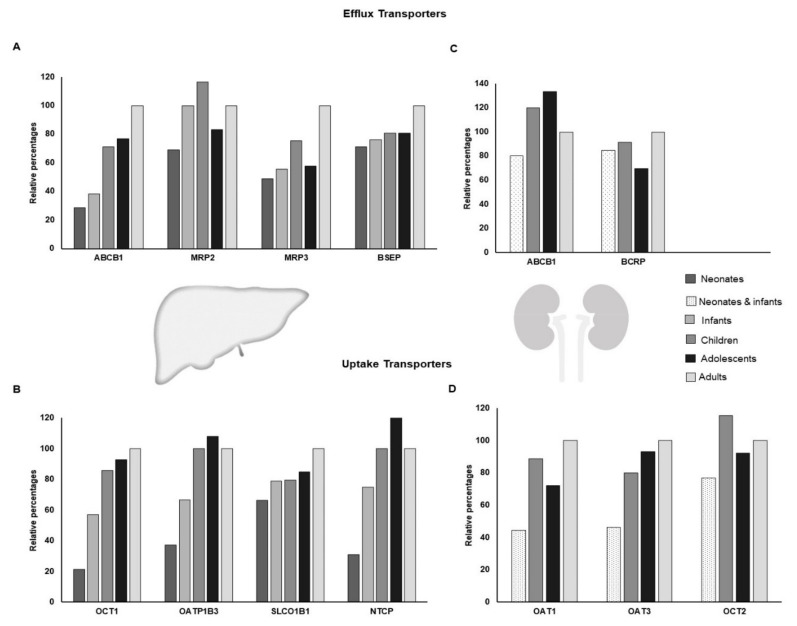
The differential expression of some of the key hepatic (**A**,**B**), and renal (**C**,**D**) transporter proteins. The *y*-axis of the bar chart denotes percentage of protein expression in comparison to adult levels. The data were combined and adopted from Prasad et al., 2016 and Cheung et al., 2019 [33,34].

**Figure 3 pharmaceuticals-15-00990-f003:**
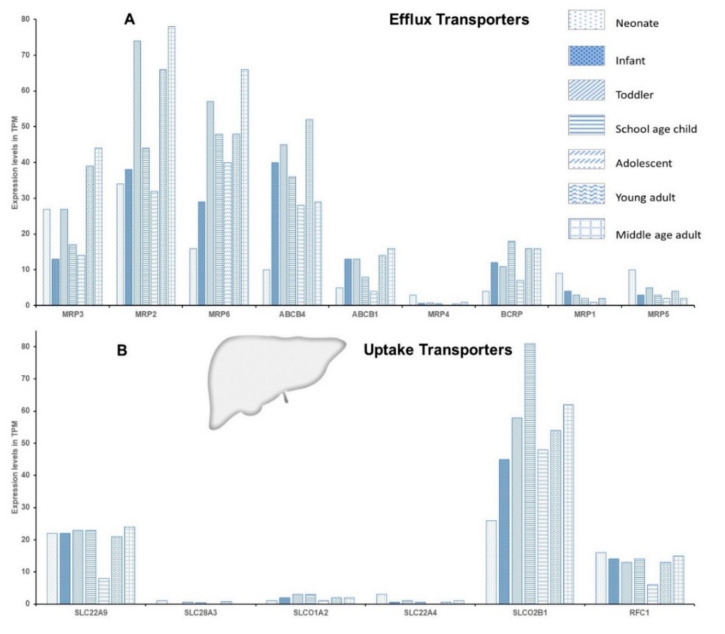
The mRNA expression of some of the key hepatic efflux (**A**) and uptake (**B**) transporters. The *y*-axis denotes expression levels in transcript per million (TPM). The data were collated and adopted from EMBL-EBI Available online: https://www.ebi.ac.uk/gxa/home (accessed on 27 June 2022) [28].

**Figure 4 pharmaceuticals-15-00990-f004:**
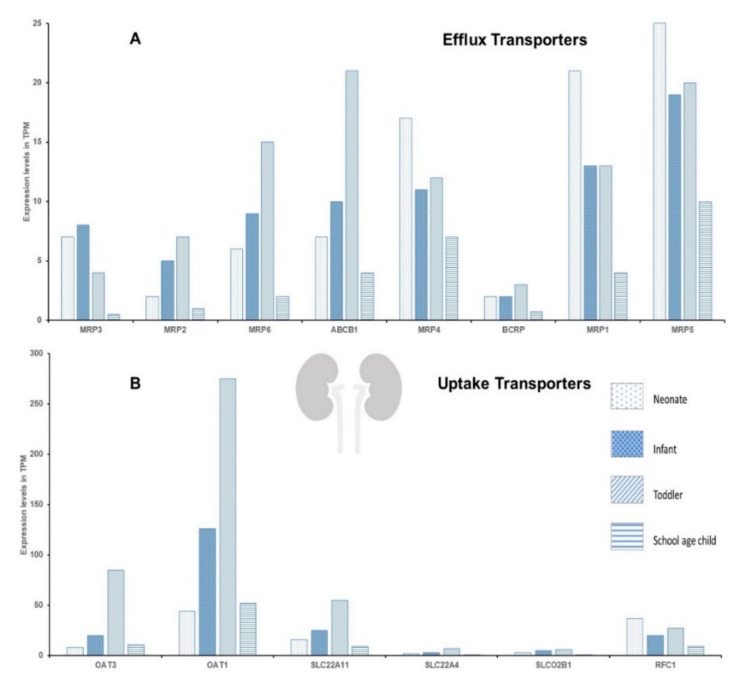
The mRNA expression of some of the key renal efflux (**A**) and uptake (**B**) transporters. The *y*-axis denotes expression levels in transcript per million (TPM). The data were collated and adopted from EMBL-EBI Available online: https://www.ebi.ac.uk/gxa/home (accessed on 27 June 2022) [28].

**Table 1 pharmaceuticals-15-00990-t001:** The anticancer agents commonly used in pediatric oncology and their drug transporters.

Mechanism of Action	Drug Name	Labelled Indications	Important Adverse Effects	Uptake Transporter/s	Efflux Transporter/s	References
Alkylating drug	Cyclophosphamide	ALL	Cytopenia, hemorrhagic cystitis, cardiotoxicity, hepatic veno-occlusive disease	NA	MRP4, MDR1, MRP2	[50,51,52]
Topoisomerase inhibitors	Doxorubicin hydrochloride	WTCKC	Cardiomyopathy, myelosuppression, secondary malignancy	NA	MDR1, MDR3, MRP1, MRP2, MRP5 BRCP	[53,54,55,56,57,58]
	Daunorubicin hydrochloride	ALL	Myocardial toxicity, myelosuppression	OATP1B1	MRP2–6	[59,60]
Enzymes	Asparaginase Erwinia chrysanthemi, Calaspargase Pegol-mknl, Pegaspargase	ALL	Hemorrhagic or thrombotic events; pancreatitis; hypersensitivity reaction; diabetic ketoacidosis; posterior reversible encephalopathy syndrome	NA	NA	[61]
Transcription inhibitor	Dactinomycin	RMS, WTCKC, NHL, ES	Veno-occlusive disease, myelosuppression, secondary malignancy	OAT4, PEPT2	MDR1	[62,63,64]
Bispecific CD19-directed CD3 T-cell engager	Blinatumomab	ALL	Cytokine release syndrome, neurological toxicity	NA	NA	[65]
CD19-directed genetically modified autologous T-cell immunotherapy	Tisagenlecleucel	ALL	Cytokine release syndrome, neurological toxicity	NA	NA	[66]
CD33-directed antibody-drug conjugate	Gemtuzumab ozogamicin	AML	Hepatotoxicity, infusion-related reactions, thrombocytopenia, neutropenia	NA	NA	[67,68,69]
Folate antagonist	Methotrexate Sodium	ALL	Bone marrow suppression, impaired renal function, hepatotoxicity, penumonitis	RFC1, OATP1B1, OATP1A2, OAT1, OAT3	MDR1, BRCP, MRP1–5, MRP7	[70,71,72,73,74,75]
GD2-binding monoclonal antibody	Dinutuximab	NB	Infusion reactions, neuropathy	NA	NA	[76]
	Naxitamab-gqgk	NB	Infusion reactions, neurotoxicity	NA	NA	[61]
Human cytotoxic T-lymphocyte antigen 4 (CTLA-4)-blocking antibody	Ipilimumab	Melanoma, CRC	Immune-mediated adverse reactions	NA	NA	[77]
Microtubule inhibitor	Vincristine Sulfate	ALL, AML, NB, NHL, RMS, WTCKC	Neuropathy, hepatic veno-occlusive disease	NA	MDR1, MRP1–3, MRP7, RLIP1	[78,79,80,81,82,83]
Kinase inhibitors	Dasatinib	ALL, CML	Myelosuppression, hemorrhage, fluid retention, cardiac dysfunction	NA	MDR1, BRCP	[84,85]
	Imatinib Mesylate	ALL, CML	Cytopenias, congestive heart failure, hepatotoxicity, hemorrhage	OCT1	MDR1	[86,87,88,89]
	Nilotinib	CML	Myelosuppression, QT prolongation, electrolyte abnormalities, pancreatitis, hepatotoxicity	NA	MDR1, BRCP, MRP6	[84,90,91]
	Crizotinib	NHL	Ocular toxicity, hepatotoxicity, interstitial lung disease	OATPB1/3, OATP2B1	MDR1	[92,93]
	Entrectinib	ST	Congestive heart failure, CNS adverse effects, fracture, hepatotoxicity, QT prolongation, vision disorders	NA	MDR1, BRCP	[94]
	Larotrectinib Sulfate	ST	Neurotoxicity, hepatotoxicity	OATP1	MDR1, BRCP	[95]
	Selumetinib Sulfate	NF Type 1	Cardiomyopathy, ocular toxicity, skin rash, diarrhoea, rhabdomyolysis	NA	MDR1, BRCP1	[96,97]
	Everolimus	GCA	Pneumonitis, infections	NA	MDR1	[98]
Programmed death ligand-1 (PD-L1) blocking antibody	Avelumab	MCC	Immune-mediated reactions	NA	NA	[61]
Programmed death receptor-1 (PD-1)-blocking antibody	Pembrolizumab	ST, HL, MCC, NHL	Immune-mediated reactions	NA	NA	[61]
	Nivolumab	CRC	Immune-mediated reactions	NA	NA	[61]
Purine nucleoside metabolic inhibitor	Clofarabine	ALL	Bone marrow suppression, infectious complications, tumor lysis syndrome, systemic inflammatory response syndrome	OAT1, OAT3, OCT1	BRCP, MRP4, MRP5	[99,100]
	Mercaptopurine	ALL	Bone marrow suppression, immunosuppression, hepatotoxicity	CNT2, CNT3, ENT1	MRP4, MRP5, MDR1	[95,101,102]
	Nelarabine	ALL, NHL	Myelosuppression, neurological toxicity	ENT1, ENT2	NA	[103,104]
Pyrimidine nucleoside metabolic inhibitor	Cytarabine	ALL, CML	Bone marrow suppression, cytarabine syndrome, cerebral and cerebellar dysfunction, bowel necrosis, pulmonary edema, cardiomyopathy	ENT1, OCTN1, CNT3	MRP1, MRP3, MRP4, MDR1	[105,106,107,108,109,110]

ALL, acute lymphocytic leukemia; AML, acute myeloid leukemia; CML—chronic myeloid leukemia; CRC—colorectal cancer; ES—Ewings sarcoma; GCA, giant cell astrocytoma; HL—Hodgkin lymphoma; MCC, Merkel cell carcinoma; NA—not available; NB, neuroblastoma; NF—neurofibromatosis; NHL—non-Hodgkin lymphoma; RMS, rhabdomyosarcoma; ST—solid tumors; WTCKC, Wilms tumor and other childhood kidney cancers.

## Data Availability

Not applicable.

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
