# Peer review of "Role of Drug Transporters in Elucidating Inter-Individual Variability in Pediatric Chemotherapy-Related Toxicities and Response"

_pharmaceuticals, 2022, doi:10.3390/ph15080990_

Round 1

Reviewer 1 Report

This manuscript provides an overview of ontogeny of drug transporters especially related to pediatrics. It is generally well written and covers the area well. There are other reviews on the subject, but an update here will be of value to the pharmaceutics community. I can recommend its publication with minor revisions.

It will be good for the reader to be introduced to DTS and the relevant proteins in developmental stages in a Figure which will improve comprehension. May be Figure 2 could be rearranged to come in front and discussed more in detail in an introductory manner.

Supplementary Figures (Page 2, line 85) in a review do not really make sense. If they are relevant, these should be merged with the main text.

Section 4 does not really need a subsection, as there is only type of enzyme discussed here.

The weakest part is section 14, and it does not really shed much light on personalized chemotherapy in pediatrics. I can understand that there are limited studies, and the authors use only one example per se to elaborate this whole section. This should be merged with conclusions and future perspective.

Author Response

Reviewer 1

  1. Comments and Suggestions for Authors; This manuscript provides an overview of ontogeny of drug transporters especially related to pediatrics. It is generally well written and covers the area well. There are other reviews on the subject, but an update here will be of value to the pharmaceutics community. I can recommend its publication with minor revisions.

Author response: Thank you for the remark and appreciation. In this manuscript, we attempted to provide a summary on the role of drug transporters in pediatric oncology setting that is more specific compared to other general reviews on this topic.

  1. It will be good for the reader to be introduced to DTS and the relevant proteins in developmental stages in a Figure, which will improve comprehension.

Author response: Within the manuscript in Figure 1 and in supplementary material, we provide such information. However, as reviewer pointed out it is not clear for the reader. As per the suggestion, we revised the manuscript with supplementary figures inserted within the main text. The developmental stages of DTS are shown in the revised manuscript in Figures 2, 3 and 4.

  1. May be Figure 2 could be rearranged to come in front and discussed more in detail in an introductory manner.

Author response: We revised the introduction (section 1) considering the suggestion. The “Factors determining expression and function of drug transporters in pediatric chemotherapy” is relabeled as Figure 1. as advised.

  1. Supplementary Figures (Page 2, line 85) in a review do not really make sense. If they are relevant, they should be merged with the main text.

Author response: The supplementary figures are modified and inserted within the text as advised (Figures 3 and 4 in revised manuscript)

  1. Section 4 does not really need a subsection, as there is only type of enzyme discussed here.

Author response: we agree with the reviewer and modified as suggested in revised manuscript.

  1. The weakest part is section 14, and it does not really shed much light on personalized chemotherapy in pediatrics. I understand that there are limited studies, and the authors use only one example per se to elaborate on this whole section. This should be combined with conclusions and future perspectives.

Author response: As suggested, Section 14 has been revised.

Reviewer 2 Report

I have reviewed the manuscript "Role of drug transporters in elucidating inter- individual variability in pediatric chemotherapy-related toxicities and response". The manuscript is well prepared and has very critical information for clinicians working in the oncology/hematology area to select the suitable chemotherapy for pediatrics.  

Author Response

  1. I have reviewed the manuscript "Role of drug transporters in elucidating inter- individual variability in pediatric chemotherapy-related toxicities and response". The manuscript is well prepared and has very critical information for clinicians working in the oncology/hematology area to select the suitable chemotherapy for pediatrics.  

Author response: Thank you for the time and feedback